# Using Audit to Improve End-of-Life Care in a Tertiary Cancer Centre [note 1]

**DOI:** 10.3390/curroncol32080430

**Published:** 2025-07-30

**Authors:** Conor D. Moloney, Hailey K. Carroll, Elaine Cunningham, Daniel Nuzum, Mairead Lyons, Richard M. Bambury, Dearbhaile C. Collins, Roisín M. Connolly, Paula O’Donovan, Renelyn Sumugat, Shahid Iqbal, Sinead A. Noonan, Derek G. Power, Aoife C. Lowney, Seamus O’Reilly, Mary Jane O’Leary

**Affiliations:** 1Cork University Hospital, Wilton, T12 DC4A Cork, Irelanddaniel.nuzum@ucc.ie (D.N.); mairead.lyons5@hse.ie (M.L.);; 2Marymount University Hospital and Hospice, T12 A710 Cork, Irelanddralowney@marymount.ie (A.C.L.); drmjoleary@marymount.ie (M.J.O.); 3Cancer Research@UCC, University College Cork, T12 XF62 Cork, Ireland

**Keywords:** oncology, palliative care, end-of-life care (EoLC), quality improvement, audit, cancer

## Abstract

Many people with cancer die in hospitals, so it is important that they receive the best possible care at the end of life. In 2021, we found several problems in how dying patients were cared for in our hospital, including poor communication, lack of emotional or spiritual support, and limited documentation. To address this, we introduced a care planning form, a checklist, and staff training to improve how we support dying patients and their families. This follow-up study looked at whether these changes led to a measurable improvement. We found that care improved in domains such as communication, symptom control, and referral for spiritual support. The simple use of a care checklist was linked to higher quality ratings for end-of-life care. Our findings show that small, low-cost interventions and regular review can make a big difference in how patients are supported at the end of life. This model could help other hospitals improve care, too.

## 1. Introduction

Cancer is the leading cause of death in Ireland, accounting for 29% of all deaths in 2023 [1]. Many of these deaths occur in an acute hospital setting where increasing interventionalist treatment options can distract from end-of-life care (EoLC). EoLC refers to the care provided to patients in the final days to weeks of life, encompassing symptom control, communication, emotional support, and care planning. It should therefore be a core priority for oncology teams, with multidisciplinary palliative care support [2,3,4], to deliver high-quality, person-centred care to patients in the last days of their lives. Despite this, numerous studies internationally, including our audit, have highlighted inadequacies in the care experienced by patients and their families towards the end of life [5,6,7,8,9,10,11,12].

Clinical audit is the cornerstone of quality improvement in healthcare, providing a systematic method to evaluate and enhance care delivery. In the context of palliative care, audits are particularly valuable as they help identify gaps in symptom control, communication, and patient-centred care, all of which profoundly impact patients and families. While not yet universally routine, auditing in palliative care is increasingly recognised as essential to improving outcomes and guiding service development.

Pandemics expose vulnerabilities and magnify inequalities. The COVID-19 pandemic and a concurrent cyberattack led to significant challenges in the provision of cancer care (including EoLC) in Ireland [13]. These concerns led to an initial audit examining the quality of EoLC for all patients who died under our care in 2021 (*n* = 66). Deficits identified included inadequate communication with families, limited offering of pastoral care support, and lack of exploration of what was important for patients at EoL [5]. Our results were concordant with the findings of other studies [14,15], including the ‘Time to Reflect Survey’ by the Irish Hospice Foundation, in which 64% of 2200 respondents reported that their experience of the death of a loved one was impacted by the pandemic [16].

In response to our initial audit in July 2022, we introduced a care of dying patients proforma (Appendix A, Figure A1), an EoLC co-ordinator and quality checklist, and expanded our EoL committee to include Non-Consultant Hospital Doctors (NCHDs). Didactic staff education sessions for medical oncology NCHDs and nursing staff on our inpatient ward were provided by our specialist palliative care team on topics related to EoLC. This report outlines the findings of this repeat audit, which aimed to evaluate whether the introduction of structured tools and staff education improved the quality of EOLC for patients in a tertiary cancer centre.

## 2. Materials and Methods

This was a retrospective re-audit performed at a tertiary cancer centre, one of eight National Cancer Control Programme-affiliated cancer centres and a European Society of Medical Oncology (ESMO)-designated centre of Integrated Oncology and Palliative Care. Inpatient cancer care is provided on a 33-bed ward with five single, one 4-bed, and four 6-bed areas. Specialist palliative care is provided by a consultant-led team, with a registrar and clinical nurse specialists. The Hospital EoL Committee, with representation from medical oncology, palliative care, nursing, pastoral care, and hospital administration, meet monthly to provide oversight for matters involving EoLC and has responsibility for re-audit through our EoLC coordinator.

We identified all patients who died while hospitalised under the care of the medical oncology service between 11 July 2022 and 30 April 2023. Patients’ physical and electronic records were reviewed. Data on the quality of EoLC delivered during their final hospitalisation were collected and managed using Microsoft Excel. Auditing was conducted by a single trained auditor (C.D.M.), with oversight from the EoLC coordinator and the hospital’s EoL committee.

Quality of EoLC was assessed using the Oxford Quality indicators for mortality review (Figure 1) [17], a tool based on UK National Audit of Care at the End of Life audit tools and designed for routine mortality review in clinical practice. It comprises five domains: recognising the possibility of imminent death, communication with the dying person, communication with families and others, involvement in the decision making, and individualised plan of care. Lack of specific documentation in the patient chart was interpreted as absence of a domain, in keeping with standard audit methodology. No significant missing data was observed. An overall score for EoLC was assigned on a numerical scale from 1 (very poor) to 5 (excellent). This tool was chosen for the initial audit as it is designed for routine mortality review in clinical practice.

Patient demographics, the overall quality of EoLC, and the individual domains of EoLC were compared to the initial audit cohort. Chi-squared tests were used to determine statistical significance for categorical variables. Statistical analysis was performed using SPSS v29. Ethical approval was granted by the Hospital Quality and Patient Safety Department (approval number CUH-AUD-2022/018). Informed consent was not required as this was a retrospective medical chart review. Preliminary findings from this audit were previously presented in part as a poster at the European Society of Medical Oncology Congress 2023 (Poster 1597) [18].

## 3. Results

We identified 72 patients (41 female, 31 male) who died under the care of the medical oncology service. The median age at death was 65 years [range 23–89]. The median length of an admission resulting in death was 12 days [range 0–118]. A total of 62 deaths occurred on inpatient wards, 6 in the emergency department, and 4 in the intensive care or high-dependency units.

The risk of imminent death was documented for 68 patients (94.4%). This risk was communicated to 53 patients (73.6%) and 63 of their families (87.5%), increases from the prior audit of 17.7% and 4.7%, respectively. “Do not attempt cardiopulmonary resuscitation” (DNACPR) orders were completed for 66 patients (91.6%). Unnecessary interventions and investigations were discontinued appropriately in 59 cases (81.9%).

Symptom assessments were documented for 65 patients (90.2%). A total of 58 patients (80.5%) received specialist inpatient palliative care input. Exploration of what was important to the patient and their family was noted in 33 cases (45.8%) and pastoral care or faith advisor input was offered in 42 (58.3%), numerical increases of 21.6% and 47.7%, respectively. All domains, except documentation of risk of imminent death, demonstrated a numerical increase following our interventions (see Figure 2). No significant missing data were observed for key domains.

The mean quality of EoLC was very good (mean = 4.0). A chi-square test showed a statistically significant improvement in EoLC quality scores in the re-audit compared to the initial audit (χ^2^ (3, *n* = 138) = 9.75, *p* = 0.021). The strength of association was small to moderate (Cramér’s V = 0.266). The most frequently received level of care was excellent (i.e., the mode was 5, excellent, *n* = 28). No patient was assessed to have received very poor (score = 1) EoLC, and the proportion of patients receiving poor (score = 2) EOLC decreased (8.3% vs. 21.2%). The care of dying patients guidance proforma was employed in 42 cases (58.3%). The use of the EoLC proforma was associated with significantly higher quality scores, χ^2^ (3, *n* = 70) = 40.21, *p* < 0.001, indicating a strong positive impact (Cramér’s V = 0.758). Overall, the results recorded using the Oxford scale of 1–5 were as follows: very poor (1): 0, poor (2): 6, satisfactory (3): 15, good (4): 23, and excellent (5): 28.

## 4. Discussion

This audit found significant improvements in the quality of EoLC for patients in a tertiary cancer centre following the implementation of structured interventions. Use of the care of dying patients proforma was associated with a statistically significant improvement in the quality of EoLC, highlighting it as an effective tool to improve EoLC. Our findings align with prior studies where a similar benefit to the use of proformas or simple interventions, particularly with respect to improved documentation, was observed [17,19,20,21,22,23].

Our results also further validate the use of the Oxford indicators as an audit tool for assessing quality of EoLC at mortality review. The improvement we have seen between audit cycles is consistent with the experience in Oxford University Hospitals, where 3 years of re-audit with this tool have seen scores for the EoL phase of mortality review increase year-on-year [17].

The majority of patients under our care received good or excellent EoLC. Despite numerical improvements in the proportion of patients documented to have been offered pastoral care or had what was important to them explored, these remain areas of relative weakness compared to other domains of EoLC. The role of pastoral care in improving the quality of dying is well recognised [24,25], and continued further prioritisation of this is needed.

The role of didactic education initiatives for healthcare staff in improving EoLC requires further study, particularly as we did not include an assessment of staff experiences. A 2023 systemic review by Bainbridge and colleagues noted that while education interventions can increase health practitioners’ confidence in aspects of EoLC, more objective outcomes were mixed [26]. High-quality EoLC should be everyone’s business. This means that the target audience for teaching is everyone, from frontline clinicians to ward receptionists, catering, porters, and patients and their carers. Diverse teaching and educational strategies are needed to suit these varying roles, with different requirements of knowledge and practical competency. Communication is an especially important skill for healthcare staff to develop in dealing with patients and their families at EoL, and more focused, simulation-based sessions have been shown to improve communication skills in addressing EoL subjects [27]. This may, therefore, be a more effective approach to staff education.

This study has several limitations. Data collection was performed retrospectively and was limited to what was documented in patient records. Actions such as offering pastoral care or communicating with families may occur more frequently than is seen here but lack documentation. Our study also does not capture other important outcomes, such as the experiences of patients’ families or the impact of death and dying on healthcare staff. Doctors, for example, often feel unprepared or unsupported when providing EoLC, which can cause significant emotional distress [28,29].

High-quality EoLC for patients with cancer is complex and multidisciplinary. It is therefore crucial that education and training in delivering EoLC is emphasised to improve continuing professional development and competency for staff at all levels. This study provides a template for how simple interventions, re-audit, and a multidisciplinary approach can be the first step in creating an environment where good-quality EoLC is valued. Through our EoL committee, we are committed to ongoing audit and quality improvement of EoLC. We plan to enhance our educational strategies by surveying junior doctors to tailor content.

## 5. Conclusions

This work highlights care needs and quality improvements that extend beyond disease-modifying treatments, emphasising person-centred EoLC in oncology. Simple, low-cost interventions, such as care planning proformas and structured education, can meaningfully improve EoLC in oncology. Continued commitment to audit cycles and multidisciplinary collaboration is essential. These findings are relevant not only in academic settings but also for broader clinical practice, where scalable models for improving EoLC are urgently needed.

The difficulties experienced during the COVID-19 pandemic served as the catalyst for our audits and interventions. Climate change is anticipated to make natural disasters, extreme weather events, and pandemics more common [30,31]. The COVID-19 pandemic and analysis of other natural disasters have shown that these events can have profound impacts on EoLC [32]. It is, therefore, essential that we apply our learnings here to future scenarios.

## Figures and Tables

**Figure 1 curroncol-32-00430-f001:**
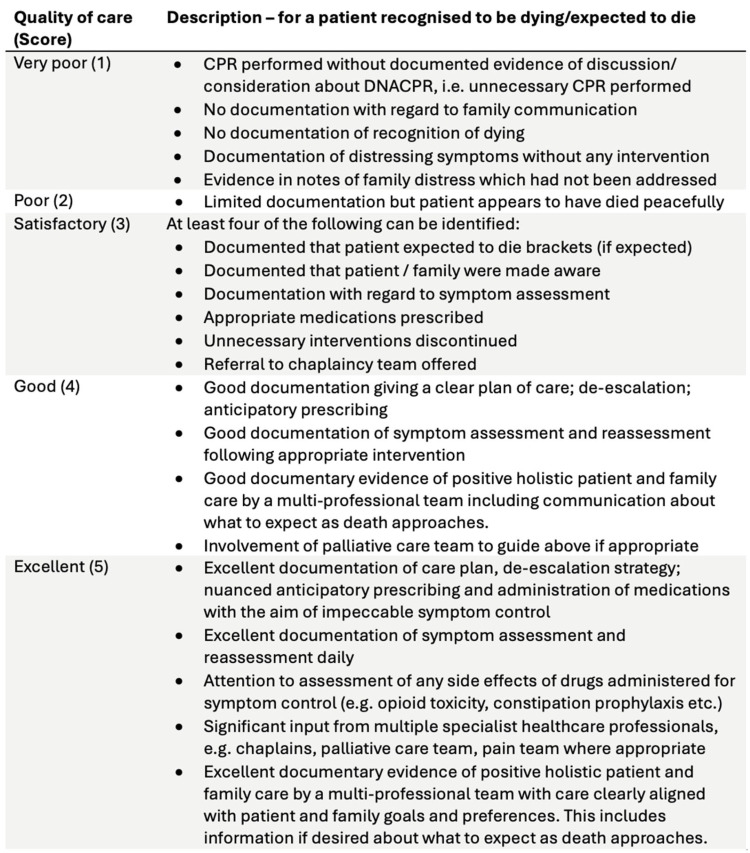
Oxford Quality indicators for mortality review.

**Figure 2 curroncol-32-00430-f002:**
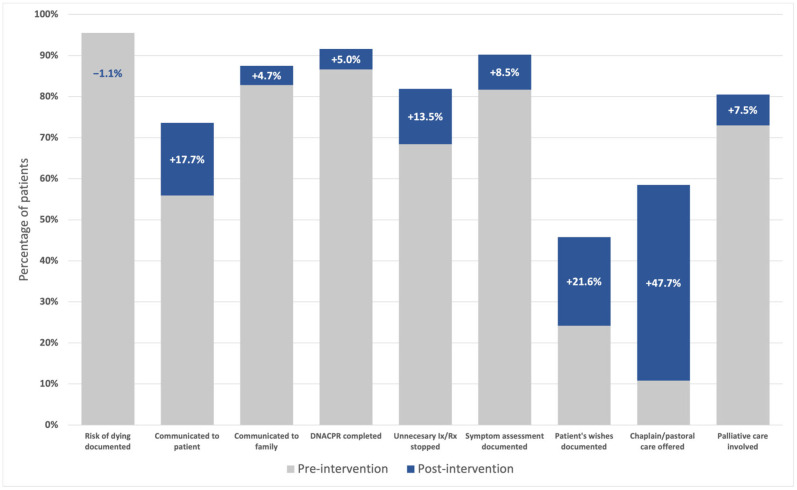
Documentation of domains of EoLC pre- and post-intervention.

## Data Availability

The original data presented in this study are openly available in FigShare at https://doi.org/10.6084/m9.figshare.29456471.v1.

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
