# Peer review of "Using Audit to Improve End-of-Life Care in a Tertiary Cancer Centre [Author-notes fn1-curroncol-32-00430]"

_curroncol, 2025, doi:10.3390/curroncol32080430_

Round 1
Reviewer 1 Report
Comments and Suggestions for Authors
Dear authors:
Congratulations on your work.
Please clarify the following points:
1) Abstract: Please review the aim of the study. I suggest "This re-audit aimed to review how these changes impacted on the care received by the patients of a Terciary Cancer Centre".
2) Introduction: Could you please speak a bit about the importance of conducting audits in the healthcare context, and then specifically in the context of palliative care? Is it a common practice? What potential benefits does auditing bring in this specific context? Also, add the aim of the study at the end of introduction.
3) Methods: Were all the patients included in the study those who were followed during the period, or only those who died while hospitalized?
4) Methods: How did you organize the audit records? In an Excel table? Was it carried out by a single auditor or a team of auditors? If it was a team, how was it composed?
5) Results: Are these results similar to any other studies? Please add National and international comparisions.
6) Conclusion is missing. It should be compreensive and address the aims established for this study. Please add suggestions regarding different contexts (academic and practice).
Best regards
Author Response
We sincerely thank the reviewer for their thoughtful comments. We have revised the manuscript in line with their suggestions and believe it has been strengthened as a result. Our point-by-point responses are included below:
Comment 1: Abstract: Please review the aim of the study. I suggest "This re-audit aimed to review how these changes impacted on the care received by the patients of a Terciary Cancer Centre".
Response 1: Thank you, the abstract aim has been reworded to “This re-audit aimed to review how these changes impacted the care received by patients in a Tertiary Cancer Centre.”
Comment 2: Introduction: Could you please speak a bit about the importance of conducting audits in the healthcare context, and then specifically in the context of palliative care? Is it a common practice? What potential benefits does auditing bring in this specific context? Also, add the aim of the study at the end of introduction.
Response 2: We added a new paragraph explaining the role of audit in healthcare and its specific value in palliative care. We concluded the introduction with a clearly stated aim of the study.
Comment 3: Methods: Were all the patients included in the study those who were followed during the period, or only those who died while hospitalized?
Response 3: We have clarified that only patients who died while hospitalised under the care of the medical oncology service were included.
Comment 4: Methods: How did you organize the audit records? In an Excel table? Was it carried out by a single auditor or a team of auditors? If it was a team, how was it composed?
Response 4: We have included that data were collected using Microsoft Excel and reviewed by a single trained auditor (C.M.), with oversight from the End-of-Life Care (EoLC) coordinator and committee.
Comment 5: Results: Are these results similar to any other studies? Please add National and international comparisions.
Response 5: We have added comparisons with other studies in the Discussion section, including national and international audit findings, such as those from Oxford University Hospitals.
Comment 6: Conclusion is missing. It should be compreensive and address the aims established for this study. Please add suggestions regarding different contexts (academic and practice).
Response 6: Many thanks for this suggestion. A conclusion section has been added to summarise the findings, their relevance to both academic and clinical practice, and the importance of interdisciplinary collaboration.
Reviewer 2 Report
Comments and Suggestions for Authors
This paper discusses the results of an audit conducted at the Tertiary Cancer Centre to determine if there has been an improvement in the quality of End-of-Life Care (EoLC).
The paper describes the benefits of simple interventions, the importance of re-audit, and the role of ongoing interdisciplinary commitment.
This paper adds a small new insight into EoLC in patients with cancer.
If you revise the paper based on the following 16 comments, this paper will improve.
Major comments are 6, 7, 8, 14, and 15.
Minor comments are other than the above.
1. The title includes “Beyond Chemotherapy”. However, the abstract and text do not contain sufficient content. Please make sure that the title matches the abstract and the text.
2. The phrase “End-of-Life Care” is included in the title and elsewhere. Please provide an operative definition of “End-of-Life Care” where appropriate.
3. A simple summary is desirable, but it is not desirable to be stated ambiguously. Please describe “a difference” in line 18, “area assessed” in line 19, and “better outcome” in line 20, using specific, unambiguous, simple language.
4. In the first sentence of your abstract, please briefly describe the rationale for writing this paper. In the second sentence, please briefly describe the research gap.
5. The abstract states "a care of dying patients proforma, EoLC quality checklist, targeted education and training for staff, and an expanded end-of-life (EoL) committee". However, the order in which the items are listed in the Simple Summary and Text differs. For example, in the Simple Summary, the phrase “a checklist, care planning form, and staff training” is written, but we suggest that it be revised to "care planning form, a checklist, and staff training. Please modify the order in which they are listed in the Simple Summary, Abstract, and Text.
6. The abstract, on lines 31-32, states “The mean quality of EoLC was 4.0, compared to 3.5 pre-intervention”. Please describe whether there is a statistically significant difference or not. Please describe the P-value, effect size, and 95% confidence interval if possible.
7. Please add to the last paragraph in the introduction section about the research gap in this paper.
8. In the Materials and Methods section, line 84, please add a citation for “Oxford Quality indicators”.
9. In the Materials and Methods section, line 97, concerning “Ethical approval,” provide the approval number.
10. In the Materials and Methods section, please add whether or not you provided informed consent to the 72 patients or how you ensured that the 72 patients had the right to refuse the use of their data for the study.
11. In the Materials and Methods section, please describe how you processed the missing data.
12. In the Results section, on line 100, 72 patients (41 female) are listed. Please add the number of male patients.
13. Please add in the Results section whether or not there is any missing data.
14. In the Results section, lines 121-122, the following is noted.
Average quality of EoLC was very good (mean = 4.0). This is a numerical increase from 3.5 pre-intervention.
If this is the primary result, please state the P-value, effect size, and 95% confidence interval. For other results, please provide P-value, effect size, and 95% confidence interval for 2-arm comparisons.
15. In the first paragraph of the Discussion section, please include a summary related to the key findings of this paper.
16. In the Discussion section, lines 160-166 contain discussion related to COVID-19. However, this does not reflect the Results section.
Please remove lines 160-166 from the Discussion section, or add content related to COVID-19 in the Results section.
Based on the reviewer's comments, please revise or provide any objections to the comments. Objectives of opinions are welcome. However, reviewers' comments should not be ignored.
Please highlight and revise each of the reviewers' comments by using methods such as making the text red or the background of the text yellow.
Author Response
We sincerely thank the reviewer for their thoughtful comments. We have revised the manuscript in line with their suggestions and believe it has been strengthened as a result. Our point-by-point responses are included below:
Comment 1: The title includes “Beyond Chemotherapy”. However, the abstract and text do not contain sufficient content. Please make sure that the title matches the abstract and the text.
Response 1: We have revised the Discussion and Conclusion to explicitly highlight how the findings reflect improvements in supportive and person-centred care that extend beyond disease-modifying treatment.
Comment 2: The phrase “End-of-Life Care” is included in the title and elsewhere. Please provide an operative definition of “End-of-Life Care” where appropriate.
Response 2: A definition of EoLC has been added to the Introduction: “end-of-life care (EoLC) refers to the care provided to patients in the final days to weeks of life, encompassing symptom control, communication, emotional support, and care planning.”
Comment 3: A simple summary is desirable, but it is not desirable to be stated ambiguously. Please describe “a difference” in line 18, “area assessed” in line 19, and “better outcome” in line 20, using specific, unambiguous, simple language.
Response 3: Thank you, we have revised the vague phrasing for clarity.
Comment 4: In the first sentence of your abstract, please briefly describe the rationale for writing this paper. In the second sentence, please briefly describe the research gap.
Response 4: We added two introductory sentences: “High-quality end-of-life care is a critical yet often underemphasised component of oncology. During the COVID-19 pandemic, systemic deficits in EoLC delivery were exposed, highlighting the need for structured quality improvement.”
Comment 5: The abstract states "a care of dying patients proforma, EoLC quality checklist, targeted education and training for staff, and an expanded end-of-life (EoL) committee". However, the order in which the items are listed in the Simple Summary and Text differs. For example, in the Simple Summary, the phrase “a checklist, care planning form, and staff training” is written, but we suggest that it be revised to "care planning form, a checklist, and staff training. Please modify the order in which they are listed in the Simple Summary, Abstract, and Text.
Response 5: We have revised the order of interventions to “care planning form, quality checklist, staff education, and expanded EoLC committee” and ensured this was consistent.
Comment 6: The abstract, on lines 31-32, states “The mean quality of EoLC was 4.0, compared to 3.5 pre-intervention”. Please describe whether there is a statistically significant difference or not. Please describe the P-value, effect size, and 95% confidence interval if possible.
Response 6: We have used a more appropriate statistical analysis for assessing the differences between our initial audit and the re-audit (Chi-square test): “Quality of EoLC improved significantly when compared to the initial audit (χ²(3, N=138) = 9.75, p = 0.021).”
Comment 7: Please add to the last paragraph in the introduction section about the research gap in this paper.
Response 7: We concluded the Introduction with the following: “Despite this, numerous studies internationally, including our audit, have highlighted inadequacies in care experienced by patients, and their families, towards the end of their lives”
Comment 8: In the Materials and Methods section, line 84, please add a citation for “Oxford Quality indicators”.
Response 8: A citation has now been added, thank you.
Comment 9: In the Materials and Methods section, line 97, concerning “Ethical approval,” provide the approval number.
Response 9: Already present in the manuscript: CUH-AUD-2021/018.
Comment 10: In the Materials and Methods section, please add whether or not you provided informed consent to the 72 patients or how you ensured that the 72 patients had the right to refuse the use of their data for the study.
Response 10: We clarified that informed consent was not required because this was a retrospective review of anonymised patient records.
Comment 11: In the Materials and Methods section, please describe how you processed the missing data.
Response 11: We have stated that no significant missing data were observed. In keeping with standard audit methodology, undocumented domains were considered absent.
Comment 12: In the Results section, on line 100, 72 patients (41 female) are listed. Please add the number of male patients.
Response 12: The number of male patients has been added, thank you
Comment 13: Please add in the Results section whether or not there is any missing data.
Response 13: A line clarifying this has been added, thank you.
Comment 14: In the Results section, lines 121-122, the following is noted.
Average quality of EoLC was very good (mean = 4.0). This is a numerical increase from 3.5 pre-intervention.
If this is the primary result, please state the P-value, effect size, and 95% confidence interval. For other results, please provide P-value, effect size, and 95% confidence interval for 2-arm comparisons.
Response 14: We included chi-square statistics and effect sizes (Cramér’s V) for primary and subgroup analyses in the Results section.
Comment 15: In the first paragraph of the Discussion section, please include a summary related to the key findings of this paper.
Response 15: We added a summary sentence at the beginning of the Discussion: “This audit found significant improvements in quality of EoLC for patients in a Tertiary Cancer Centre following the implementation of structured interventions.”
Comment 16: In the Discussion section, lines 160-166 contain discussion related to COVID-19. However, this does not reflect the Results section. Please remove lines 160-166 from the Discussion section, or add content related to COVID-19 in the Results section.
Response 16: As the COVID-19 pandemic was the initial catalyst for our first audit, we felt it appropriate to retain references to the pandemic. We have moved it to the conclusion as we feel our results have relevance for future pandemic scenarios, which are anticipated to become more common with advancing climate change.
Round 2
Reviewer 1 Report
Comments and Suggestions for Authors
Dear authors:
Congratulations on the improvement made to your manuscript.
Author Response
We once again thank the reviewer for their time and consideration in reviewing our manuscript.
No changes were suggested on this occasion, but the abstract has had minor updates based on feedback from reviewer 2.
Reviewer 2 Report
Comments and Suggestions for Authors
The paper was revised based on my comments.
As a result, the paper has been improved.
However, the revisions are inadequate in the following two points.
On the other hand, I can understand the author's response.
Therefore, I make minor comments on the following two points, but do not necessarily force a revision.
Please reconsider them.
Minor Comment 1
In Peer Review Comment 1 in Round 1, I stated to you as follows.
The title includes “Beyond Chemotherapy”. However, the abstract and text do not contain sufficient content. Please make sure that the title matches the abstract and the text.
You answered, "What is Beyond Chemotherapy?
However, the abstract and text do not contain the word “Beyond Chemotherapy”.
So, I feel that your paper is difficult to understand.
However, other reviewers may have a different impression.
Please consider including the word “Beyond Chemotherapy” itself in your abstract and text.
However, this suggestion is not forced; you can disagree with my comments.
Minor Comment 2
In Peer Review Comment 10 in Round 1, I stated the following:
In the Materials and Methods section, please add whether or not you provided informed consent to the 72 patients or how you ensured that the 72 patients had the right to refuse the use of their data for the study.
You responded that informed consent was not required.
I agree with you on that point.
However, as an ethical consideration, the following wording is necessary.
E.g..
“By posting on our hospital's website that we will be conducting a retrospective review of anonymized patient records, we have ensured that patients who see the website have the right to refuse to have their data used for research.”
In my country, these ethical considerations are necessary. However, the situation may be different in different countries.
Please reconsider the above points. However, this suggestion is not forced; you can disagree with my comments.
Author Response
We thank Reviewer 2 for their thoughtful re-review of our manuscript. We sincerely appreciate your constructive input and detailed feedback throughout this process. Below we respectfully respond to your two additional comments:
Comment 1:
"In Peer Review Comment 1 in Round 1, I stated to you as follows.
The title includes “Beyond Chemotherapy”. However, the abstract and text do not contain sufficient content. Please make sure that the title matches the abstract and the text.
You answered, "What is Beyond Chemotherapy?
However, the abstract and text do not contain the word “Beyond Chemotherapy”.
So, I feel that your paper is difficult to understand.
However, other reviewers may have a different impression.
Please consider including the word “Beyond Chemotherapy” itself in your abstract and text.
However, this suggestion is not forced; you can disagree with my comments."
Response 1:
We appreciate the reviewer’s concern about consistency and clarity. After careful consideration, we have elected to remove the phrase “Beyond Chemotherapy” from the manuscript title. We made this decision to maintain consistency and coherence throughout the manuscript, given the challenges in naturally integrating this specific phrase into our abstract and main text. We believe this revision enhances clarity and ensures that our title accurately reflects the content of our manuscript.
Comment 2:
In Peer Review Comment 10 in Round 1, I stated the following:
In the Materials and Methods section, please add whether or not you provided informed consent to the 72 patients or how you ensured that the 72 patients had the right to refuse the use of their data for the study.
You responded that informed consent was not required.
I agree with you on that point.
However, as an ethical consideration, the following wording is necessary.
E.g..
“By posting on our hospital's website that we will be conducting a retrospective review of anonymized patient records, we have ensured that patients who see the website have the right to refuse to have their data used for research.”
In my country, these ethical considerations are necessary. However, the situation may be different in different countries.
Please reconsider the above points. However, this suggestion is not forced; you can disagree with my comments.
Response 2:
We sincerely appreciate the reviewer’s valuable suggestion regarding additional ethical considerations. While we acknowledge and respect this ethical practice in other international settings, our local Institutional Review Board guidelines at Cork University Hospital do not require public notification via the hospital website for retrospective audits of anonymised patient records. Therefore, we have not adjusted the manuscript to reflect website notification, though we remain mindful of this important ethical consideration and appreciate your highlighting this aspect.